# A two-hybrid antibody micropattern assay reveals specific *in cis* interactions of MHC I heavy chains at the cell surface

**Cindy Dirscherl, Zeynep Hein, Venkat Raman Ramnarayan, Catherine Jacob-Dolan, Sebastian Springer***

Department of Life Sciences and Chemistry, Jacobs University, Bremen, Germany

**Abstract** We demonstrate a two-hybrid assay based on antibody micropatterns to study protein-protein interactions at the cell surface of major histocompatibility complex class I (MHC I) proteins. Anti-tag and conformation-specific antibodies are used for individual capture of specific forms of MHC I proteins that allow for location- and conformation-specific analysis by fluorescence microscopy. The assay is used to study the *in cis* interactions of MHC I proteins at the cell surface under controlled conditions and to define the involved protein conformations. Our results show that homotypic *in cis* interactions occur exclusively between MHC I free heavy chains, and we identify the dissociation of the light chain from the MHC I protein complex as a condition for MHC I *in cis* interactions. The functional role of these MHC I protein-protein interactions at the cell surface needs further investigation. We propose future technical developments of our two-hybrid assay for further analysis of MHC I protein-protein interactions.
DOI: https://doi.org/10.7554/eLife.34150.001

## Introduction

Protein-protein interactions are difficult to investigate, especially when they involve membrane proteins under physiological conditions, specific protein conformations or subpopulations, low affinities, or defined locations in the cell. Such challenges are not usually met by yeast two-hybrid screens and co-immunoprecipitation approaches; instead, they require technically demanding methods such as fluorescence resonance energy transfer (FRET) or fluorescence correlation spectroscopy, which only work in some cases.

Recently, antibody-based capture assays on solid supports have been described that can be used in bait-prey experiments in live cells (*Löchte et al., 2014*; *Schwarzenbacher et al., 2008*; *Weghuber et al., 2010*). We now demonstrate an expansion of this concept to characterize location- and conformation-specific protein-protein interactions in a novel two-hybrid assay read out by fluorescence microscopy by using microprinted antibody patterns for the capture of bait proteins to spatially arrange them in the plasma membrane of live cells (*Dirscherl and Springer, 2017*) and to investigate their interaction with green fluorescent protein (GFP)-tagged prey proteins. The assay is universally applicable for the investigation of protein-protein interactions.

In this paper, we use this two-hybrid micropattern assay to solve the long-standing question which forms of major histocompatibility complex class I (MHC I) proteins associate laterally (*in cis*) on the plasma membrane. MHC I proteins consist of a polymorphic transmembrane heavy chain (HC), the non-covalently bound light chain beta-2 microglobulin ($\beta_2$m), and a peptide of eight to ten amino acids. Assembly of HC, $\beta_2$m, and peptide takes place in the endoplasmic reticulum (ER), followed by transport to the cell surface, where MHC I proteins present the bound peptides to T cell receptors of cytotoxic T cells; they also bind inhibitory receptors on Natural Killer cells. MHC I

*For correspondence:
s.springer@jacobs-university.de

Competing interests: The authors declare that no competing interests exist.

antigen presentation is central to the cellular immune response of jawed vertebrates against viruses, intracellular parasites, and tumors.

At the cell surface, MHC I proteins exist in three different forms: the $HC/\beta_2m$/peptide trimers, the 'peptide-empty' $HC/\beta_2m$ dimers derived from them by dissociation of the peptide, and the 'free' heavy chains derived from the dimers by dissociation of $\beta_2m$. While the lateral association of MHC I proteins has been observed before (*Capps et al., 1993*; *Arosa et al., 2007*; *Lu et al., 2012*; *Matko et al., 1994*), other researchers have not detected them (*Szöllösi et al., 1989*; *Damjanovich et al., 1983*; *Liegler et al., 1991*). Optical methods have not provided conclusive evidence which of the three forms are interacting, and biochemical approaches have not clarified where in the cell the binding occurs (*Capps et al., 1993*; *Matko et al., 1994*). To add to the complexity, the $HC/\beta_2m$ dimers and the free heavy chains have short half-lives in the cell, which complicates their analysis (*Montealegre et al., 2015*). For the study of the proposed functions of the MHC I protein-protein interactions (endocytosis, synapse architecture, inflammatory response, receptor modulation; see the discussion), knowledge of location and conformation of the associated proteins is essential (*Dixon-Salazar et al., 2014*; *Chen et al., 2017*; *Burian et al., 2016*; *Nizsalóczki et al., 2014*; *Mocsár et al., 2016*).

Our work now shows conclusively that MHC I free heavy chains, but not $HC/\beta_2m$/peptide trimers or $HC/\beta_2m$ dimers, associate *in cis* at the plasma membrane.

## Results

### H-2K$^b$ and H-2D$^b$ are specifically captured by antibody micropatterns

Membrane proteins on the surface of living cells can be captured into geometric shapes by antibodies that are printed on to the substrate in micrometer-sized patterns (*Figure 1A*; [*Dirscherl et al., 2017*]). We reasoned that any protein that naturally interacts with a captured protein would also be recruited into the patterns, and that this might be used for a protein-protein interaction assay (*Schwarzenbacher et al., 2008*). We further reasoned that if we printed antibodies that recognize only certain forms of MHC I proteins, the interaction assay might be made specific for certain forms of MHC I proteins.

We first tested whether the two common $\beta_2m$-dependent monoclonal antibodies Y3 (which binds to two forms of the murine MHC I allotype H-2K$^b$, or K$^b$ for short, namely the K$^b$HC/$\beta_2m$ dimers and K$^b$HC/$\beta_2m$/peptide trimers[*Hämmerling et al., 1982*]) and 27-11-13S (which binds to two forms of the murine MHC I allotype H-2D$^b$, namely D$^b$HC/$\beta_2m$ dimers and D$^b$HC/$\beta_2m$/peptide trimers [*Ozato and Sachs, 1981*]) were still specific for their target allotypes when used in the pattern capture assay. We inked poly(dimethylsiloxane) (PDMS) stamps with solutions of Y3 and 27-11-13S and printed them onto the surface of untreated glass coverslips. We then seeded human STF1 fibroblasts expressing C-terminal green fluorescent protein (GFP) fusions of either K$^b$ or D$^b$ onto these coverslips and observed capture of K$^b$-GFP and D$^b$-GFP by confocal laser scanning microscopy (*Figure 1B*). As anticipated, K$^b$-GFP was only captured with Y3, and D$^b$ only with 27-11-13S. We conclude that the printed $\beta_2m$-dependent antibodies still specifically recognize their target allotypes.

In addition to $\beta_2m$-dependent capture by Y3 or 27-11-13S, we wished to be able to capture MHC I proteins independently of their $\beta_2m$ or peptide association. Thus, we next tested whether MHC I proteins can also be captured *via* an N-terminal (extracellular) influenza hemagglutinin (HA) epitope tag (*Figure 1C*, bottom). We printed patterns of the monoclonal anti-HA antibody 12CA5 and seeded STF1 cells expressing either a HA-K$^b$-GFP fusion construct or K$^b$-GFP, which lacked the HA epitope. As expected, only HA-K$^b$-GFP was captured, but not K$^b$-GFP (*Figure 1D*). The HA tag did not interfere with the capture of HA-K$^b$-GFP on Y3 antibody micropatterns (*Figure 1D*). We conclude that the anti-HA antibody can be used to specifically capture HA-tagged MHC I proteins.

### Stabilizing effect of conformation-specific antibodies allows for differential patterning of K$^b$ dimers and free heavy chains

We next sought to establish conditions in which K$^b$HC/$\beta_2m$ dimers, without peptide, are preferentially captured in the antibody patterns. In STF1 cells, which lack TAP (the transporter associated with antigen processing) and thus cannot load MHC I proteins with high-affinity peptides in the ER (*de la Salle et al., 1999*), such dimers can be accumulated at the cell surface by incubation at 25°C

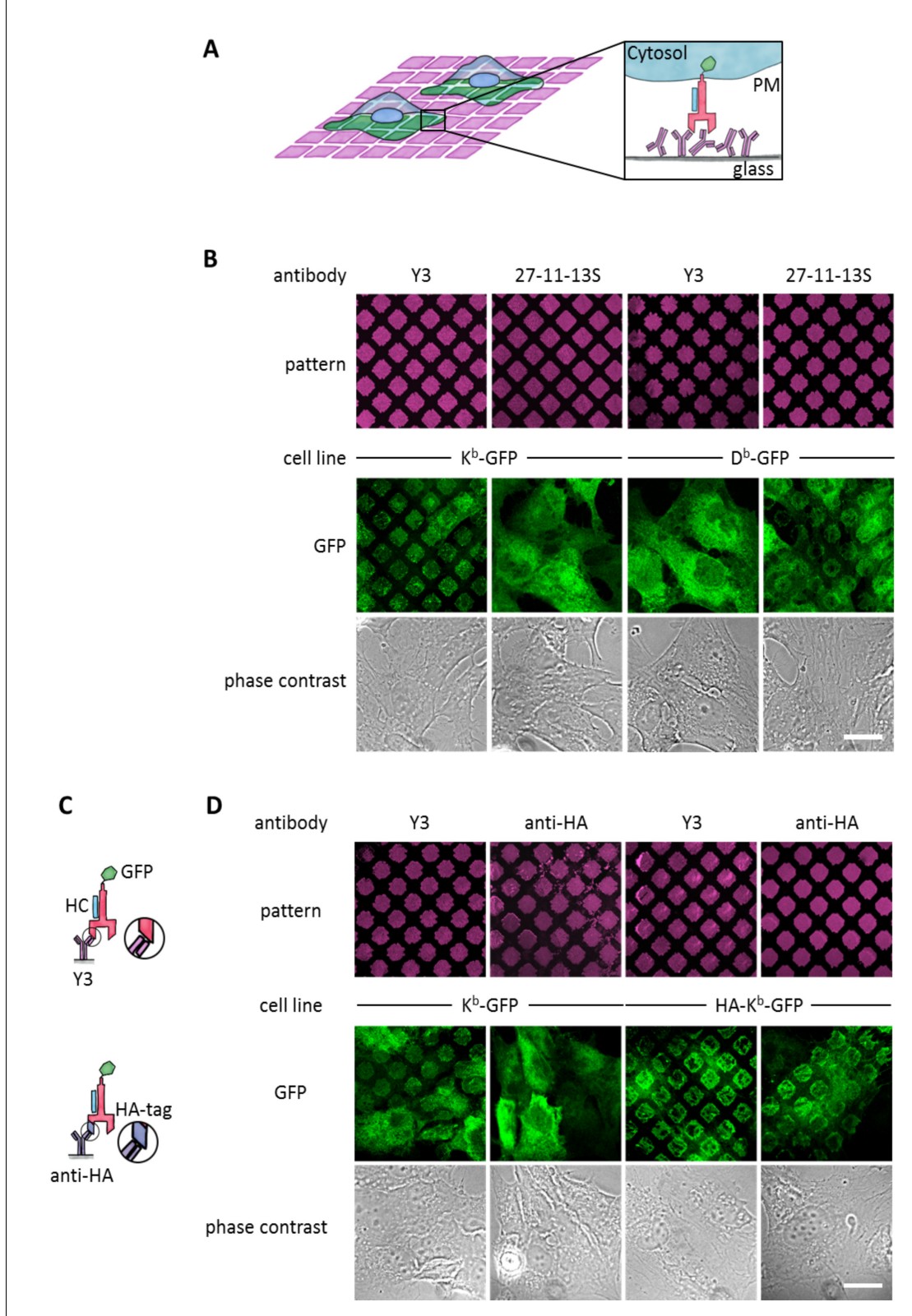

**Figure 1.** Specific capture of cell surface $K^b$ on antibody micropatterns. (**A**) Schematic presentation of the capture assay. Cells transduced with $K^b$ (red) fused to GFP (green) are incubated on the Y3 antibody micropatterns (anti $K^b$; magenta). Upon specific antibody-antigen interaction, $K^b$-GFP is captured on its extracellular epitope by the Y3 antibody pattern elements (see enlargement). (**B**) Printed antibodies are target-specific. Control experiments demonstrate that $K^b$-GFP is only captured by the anti-$K^b$ antibody Y3 and not by an antibody specific for $D^b$ (27-11-13S). (**C**) Schematic
*Figure 1 continued on next page*

*Figure 1 continued*

displaying the different antibody epitopes on the K$^b$ molecule. The Y3 epitope reacts specifically with residues of the $\alpha_2$ helix of K$^b$-GFP whereas the anti-HA antibody recognizes the additional HA-tag that was N-terminally fused to K$^b$-GFP. (D) Surface K$^b$-GFP can be directly captured by the anti-K$^b$ antibody Y3 or by the anti-HA antibody against the N-terminally tagged HA-K$^b$-GFP. Cells were transduced with K$^b$-GFP or HA-K$^b$-GFP and tested for specificity on Y3 or anti-HA antibody micropatterns. Y3 successfully captures both constructs, whereas HA only recognizes the HA-tagged molecules. Scale bar: 25 μm.

DOI: https://doi.org/10.7554/eLife.34150.002

(*Ljunggren et al., 1990*; *Montealegre et al., 2015*). These peptide-receptive K$^b$HC/$\beta_2$m dimers can be detected by subsequent binding of fluorescently labeled peptide (*Saini et al., 2013*).

We therefore printed Y3 on glass coverslips, seeded STF1 cells expressing HA-K$^b$-GFP onto the patterns, and then incubated at 25°C overnight. Next morning, we added the K$^b$-specific peptide SIINFEKL(abbreviated SL8) labeled with the TAMRA fluorophore (SL8$^{TAMRA}$). We observed a striking patterned staining of the fluorescent peptide, which demonstrates that peptide-receptive K$^b$/$\beta_2$m dimers had been captured in the patterns (*Figure 2A*, column 2). To show that binding of the peptide was specific, we pre-incubated the cells with unlabeled SIINFEKL peptide, which blocked SL8$^{TAMRA}$ binding (*Figure 2A*, column 3). Thus, we were able to capture K$^b$HC/$\beta_2$m dimers into patterns and subsequently bind peptide to them.

We then repeated the same experiment on patterns of anti-HA antibody, with the same result (*Figure 2A*, column 5). Thus, both Y3 and anti-HA antibodies captured peptide-receptive K$^b$HC/$\beta_2$m dimers on the surface of the STF1 cells.

We next wished to dissociate $\beta_2$m from the captured K$^b$HC/$\beta_2$m dimers in order to obtain patterned free heavy chains. At 37°C, dissociation of $\beta_2$m from K$^b$HC/$\beta_2$m dimers occurs rapidly, whereas at 25°C, the rate of dissociation of $\beta_2$m is significantly reduced (*Montealegre et al., 2015*; *Day et al., 1995*). Thus, we repeated the above experiments on anti-HA and Y3 patterns, but before adding the SL8$^{TAMRA}$ peptide, we shifted the cells to 37°C for two to three hours. As expected, after incubation at 37°C, HA-K$^b$-GFP patterns showed no binding of SL8$^{TAMRA}$ (*Figure 2A*, column 4), which shows that the HA-captured MHC I proteins lost their peptide-binding capacity, probably due to the dissociation of $\beta_2$m. Very interestingly, HA-K$^b$-GFP captured with Y3 retained its ability to bind peptide at 37°C (*Figure 2A*, column 1). This suggests that the $\beta_2$m-dependent Y3 antibody stabilizes the K$^b$HC/$\beta_2$m dimer complex to which it is bound by preventing the dissociation of $\beta_2$m, similar to the action of peptide (Townsend and Bodmer 1989).

To test the hypothesis that the lack of peptide binding of the K$^b$ proteins that were captured by HA and incubated at 37°C was due to the loss of $\beta_2$m, we repeated the same experiment, but instead of adding fluorescent peptide, we fixed the cells and immuno-stained with the anti-$\beta_2$m antibody BBM.1 that was directly labeled with Atto 542 (BBM.1$^{Atto542}$) (*Figure 2B*). As predicted, the Y3 micropatterns stained positive for $\beta_2$m at both temperatures (*Figure 2B*, columns 1 and 2), whereas anti-HA micropatterns stained for $\beta_2$m only at 25°C (*Figure 2B*, columns 3 and 5). When we added SIINFEKL to the cells on anti-HA-micropatterns before shifting them to 37°C, we observed that BBM.1$^{Atto542}$ staining was restored in these samples (*Figure 2B*, column 4). We conclude that captured K$^b$ free heavy chains can be generated by inducing the dissociation of $\beta_2$m from K$^b$HC/$\beta_2$m dimers captured on anti-HA patterns.

Taken together, we are able to selectively hold three different forms of K$^b$ on the surface of STF1 cells: free K$^b$ heavy chains (at 37°C on HA patterns), K$^b$HC/$\beta_2$m dimers (at 25°C on anti-HA patterns, or at 25 or 37°C on Y3 patterns), and K$^b$HC/$\beta_2$m/peptide trimers (by addition of peptide on anti-HA and Y3 patterns).

## Antibody micropatterns reveal conformation-dependent *in cis* interactions of K$^b$ free heavy chains

Since we were able to distinguish the three different forms of K$^b$ held in the patterns, we next investigated whether any of these forms associate *in cis* on the plasma membrane. For this, we designed a two-hybrid assay (*Figure 3A*): One K$^b$ construct had an N-terminal HA tag but no GFP (HA-K$^b$), the other carried a C-terminal GFP but no HA tag (K$^b$-GFP). We reasoned that HA-K$^b$ would be captured by the anti-HA antibodies, but the GFP pattern would only become detectable by microscopy if HA-K$^b$

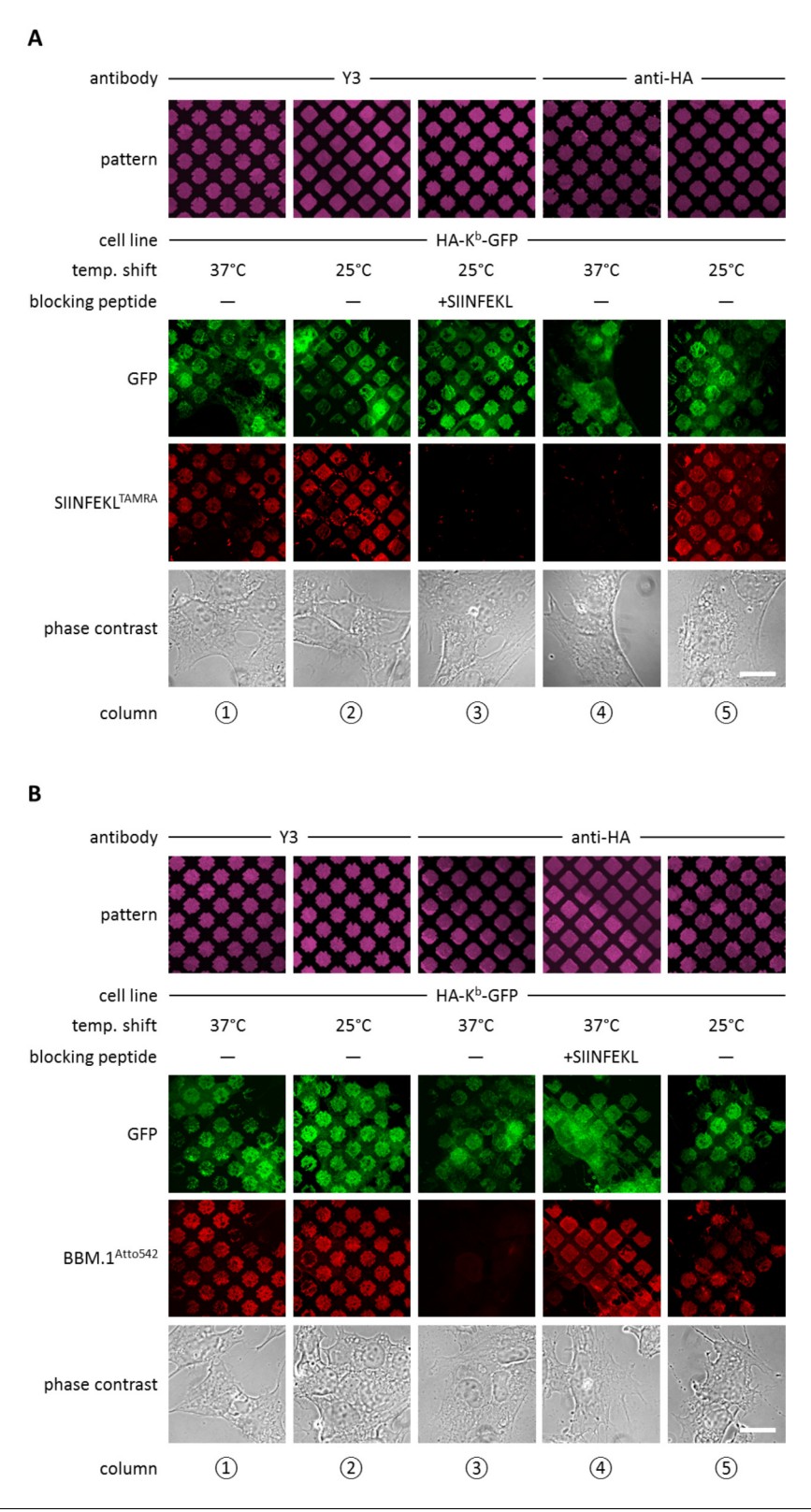

**Figure 2.** Antibody micropatterns determine stability of the captured $K^b$ population. (**A**) Cells expressing HA-$K^b$-GFP were captured on Y3 or anti-HA antibody micropatterns and incubated at 25 or 37°C to allow for the dissociation of $\beta_2$m. To identify the nature of the captured $K^b$-GFP population (green channel), specific fluorescent peptide SIINFEKL (SL8[TAMRA]; red channel) was added to the samples. Based on their ability to bind peptide, one can distinguish between the peptide-receptive $K^b$HC/$\beta_2$m dimer and the $K^b$ free heavy chains, which are incapable to bind peptide. (**B**) For further
*Figure 2 continued on next page*

*Figure 2 continued*

characterization of the captured HA-K$^b$-GFP on Y3 or anti-HA antibody micropatterns, immunostaining experiments were performed. Immunostaining of captured HA-K$^b$-GFP molecules with the anti-β$_2$m antibody (BBM.1$^{Atto542}$) reveals dissociation of β$_2$m from anti-HA antibody micropatterns at 37°C (column 3). Addition of the specific ligand peptide SIINFEKL (SL8) during 37°C incubation prevents β$_2$m dissociation (column 4). Scale bars: 25 μm.

DOI: https://doi.org/10.7554/eLife.34150.003

and K$^b$-GFP interacted together, since K$^b$-GFP alone is not captured by anti-HA micropatterns (*Figure 1D*).

To perform the experiment, we co-transduced STF1 cells with HA-K$^b$ and K$^b$-GFP, seeded the cells on anti-HA micropatterns, and incubated them overnight at 25°C to accumulate K$^b$HC/β$_2$m dimers of both K$^b$ constructs at the cell surface. The next day, we either left them at 25°C or shifted them to 37°C and followed the patterning of the GFP fluorescence.

We observed strong co-patterning of both forms after 37°C incubation, suggesting that free heavy chains can interact *in cis* in the membrane (*Figure 3B*, column 3). When we inhibited dissociation of β$_2$m by addition of SIINFEKL peptide (*Figure 3B*, column 4) or incubation at 25°C (*Figure 3B*, column 6), co-patterning was abolished. (As a control, on Y3 patterns, in contrast, where both HA-K$^b$ and K$^b$-GFP are directly bound by the antibody, strong patterning of K$^b$-GFP was visible even in the presence of SIINFEKL (*Figure 3B*, columns 1 and 2)). These data show that K$^b$ does not associate *in cis* as long as β$_2$m is bound.

To further test this conclusion, we co-transfected STF1 cells with HA-K$^b$ and scK$^b$-GFP, a single-chain construct in which the K$^b$ heavy chain and β$_2$m are linked by a peptide linker such that β$_2$m cannot dissociate (*Montealegre et al., 2015*). As in the previous control experiment, no co-patterning was observed (*Figure 3B*, column 5). These controls also demonstrated that the *in cis* interaction of the K$^b$ free heavy chains was not simply induced by the GFP domain of the K$^b$-GFP fusion proteins.

To demonstrate that the non-fluorescent HA-K$^b$ molecules were indeed present in the patterns together with K$^b$-GFP, we repeated the experiment in *Figure 3B* (column 3) with the construct E3-HA-K$^b$, in which an additional tag of a 21 amino acids (the E3 tag; see Materials and methods) is attached to the N terminus of HA-K$^b$. This tag specifically binds to the fluorescently labeled synthetic peptide, K4$^{Atto633}$. After co-patterning of K$^b$-GFP was observed, we additionally stained with K4$^{Atto633}$ and found close colocalization with K$^b$-GFP in the patterns (*Figure 3—figure supplement 1*). We conclude that the free heavy chain of E3-HA-K$^b$ is indeed captured on the patterns and then recruits the free heavy chain of K$^b$-GFP by an *in cis* interaction. We quantified this recruitment by measuring both the mean fluorescence intensity (MFI) of the entire cell and the MFI of the pattern elements in the micrographs of *Figure 3B* to standardize the increase in fluorescence signal on the pattern elements upon redistribution of K$^b$-GFP by its *in cis* interaction with HA-K$^b$ (see Materials and methods and *Figure 3C*).

Taken together, our data demonstrate that K$^b$ free heavy chains, but not K$^b$HC/β$_2$m dimers or K$^b$HC/β$_2$m/peptide trimers, associate *in cis* in the plasma membrane of live cells.

In murine (and human) cells, up to six MHC I allotypes co-exist at the cell surface. We thus wished to investigate heterotypic *in cis* interactions between free heavy chains of K$^b$ and D$^b$. In an experiment with HA-K$^b$ and D$^b$-GFP performed in analogy to *Figure 3B* above, we observed weaker but still specific (initiated by the 37°C shift) recruitment of D$^b$-GFP to the antibody pattern elements, indicating heterotypic *in cis* interactions between the two allotypes K$^b$ and D$^b$ (data not shown).

## *In cis* interactions of K$^b$ confirmed by co-immunoprecipitation

We next tested this finding in a co-immunoprecipitation experiment, without the use of micropatterns. We used the same STF1 cells co-transfected with HA-K$^b$ and K$^b$-GFP and incubated them at 25°C overnight to accumulate both K$^b$ constructs at the cell surface. The cells were then shifted to 37°C for 15 min (the half-life of K$^b$HC/β$_2$m dimers at the cell surface [*Montealegre et al., 2015*]) to dissociate β$_2$m and initiate co-localization. Then, the cells were trypsinized and lysed, and HA-K$^b$ was immunoprecipitated with the anti-HA antibody. We found efficient co-precipitation of K$^b$-GFP with HA-K$^b$, which was abolished if SIINFEKL peptide was added to the cells during the 37°C incubation (*Figure 4*). Thus, just like in the micropattern assay, the peptide clearly inhibited the interaction, suggesting that only free heavy chains were co-precipitating.

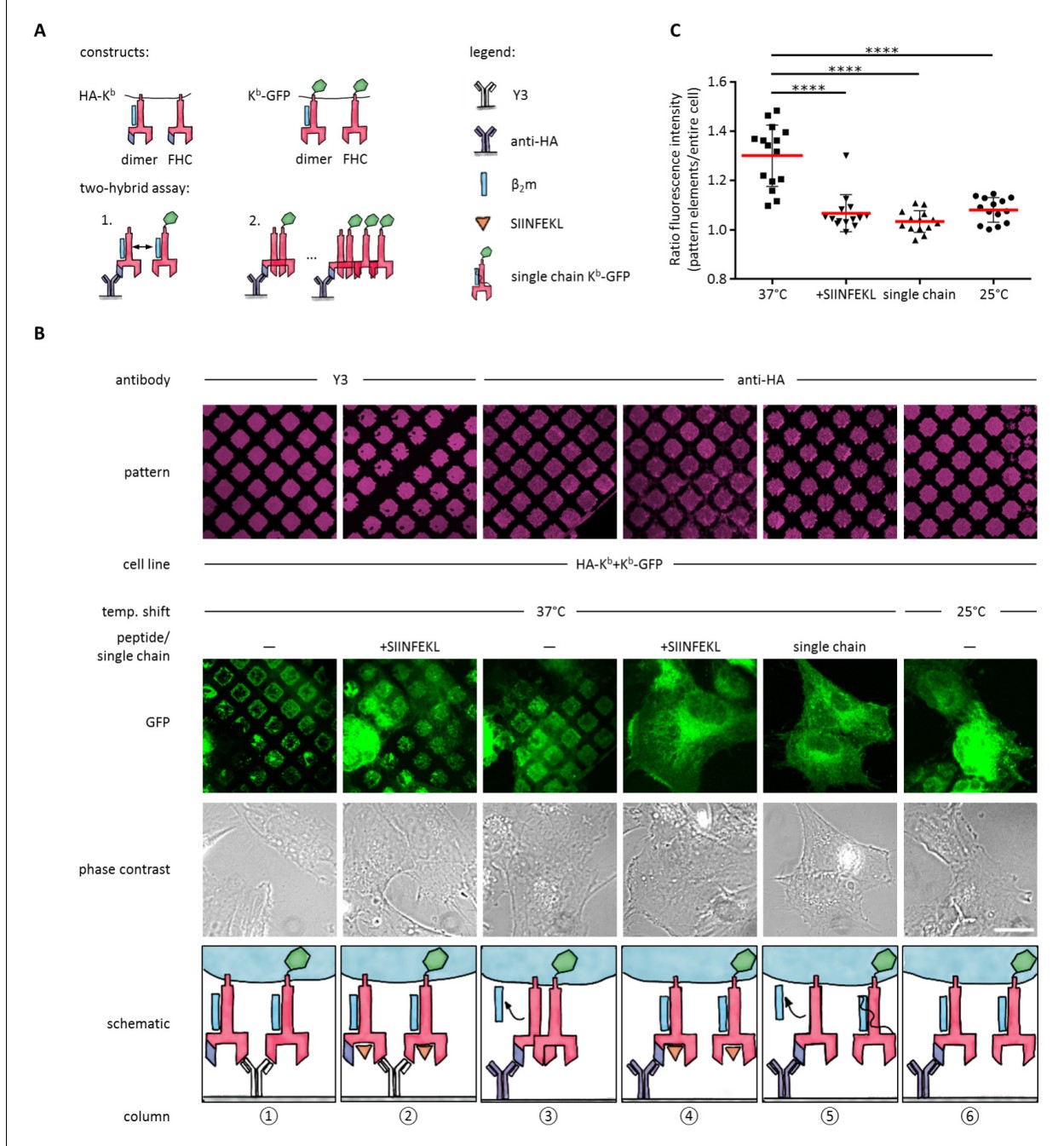

**Figure 3.** Antibody micropatterns reveal conformation-dependent *in cis* interaction of captured K$^b$-GFP. (**A**) For the two-hybrid assay, cells were co-transduced with two K$^b$ constructs: K$^b$ with an N terminal HA tag (HA-K$^b$) and K$^b$-GFP (GFP fused to the cytoplasmic tail). (**B**) Cells were incubated on anti-HA or Y3 antibody micropatterns at different temperatures. Recruitment of K$^b$-GFP (green channel) to the anti-HA antibody micropatterns occurs specifically at 37°C and can be inhibited by addition of the SIINFEKL (SL8) peptide (column 3 and 4). The single chain mutant, scK$^b$-GFP (which has β$_2$m covalently linked to the K$^b$ heavy chain) is also not recruited to the antibody micropatterns (column 5). From top to bottom: Antibody micropatterns, K$^b$-GFP, phase contrast, and schematic representation. Scale bar: 25 μm. (**C**) For quantification of co-capture, the mean fluorescence intensities of K$^b$-GFP of the total cell and the areas of pattern elements were determined. The redistribution of K$^b$-GFP leads to increased fluorescence intensity levels in the areas of the pattern elements and is represented as an increase of the ratio of the fluorescence intensity of the pattern elements over the fluorescence intensity of the entire cell (see Materials and methods). The plot shows the mean (red) ± SEM and the distribution of the calculated ratios from individual cells (black symbols; n (cells) ≥ 14) of ≥ 2 independent experiments (****: Significant difference, p<0.0001, two-tailed t-test).

DOI: https://doi.org/10.7554/eLife.34150.004

The following source data and figure supplements are available for figure 3:

*Figure 3 continued on next page*

*Figure 3 continued*

**Source data 1.** Mean fluorescence intensities for quantification of cluster formation.
DOI: https://doi.org/10.7554/eLife.34150.007
**Figure supplement 1.** Staining E3-HA-K$^b$ with K4-peptide.
DOI: https://doi.org/10.7554/eLife.34150.005
**Figure supplement 2.** Binding SIINFEKL$^{TAMRA}$ to co-captured K$^b$-.
DOI: https://doi.org/10.7554/eLife.34150.006

## Discussion

We have developed antibody micropatterns into a novel two-hybrid assay for the detailed investigation of conformation-specific *in cis* interactions of our model protein, MHC I. The versatility of our assay with extracellular HA and intracellular GFP fusions supports a broad range of applications, especially in the study of specific protein-protein interactions that require investigation in the native environment of live cells. This two-hybrid assay circumvents the disadvantages of employing two different fluorescent proteins on the cytoplasmic tails of the proteins of interest (for example for FRET microscopy) that may be biased through unspecific interactions and aggregation of the fluorescent tags.

The example of MHC I proteins demonstrates the challenges of the functional analysis of protein-protein interactions and the limitations of conventional methods, which yield no information on the spatial resolution or the distinction of different protein conformations. Previous experiments with FRET have revealed cluster formation of antibody-labelled MHC I proteins at the cells surface, but the involvement of free heavy chains was only indirectly shown (*Matko et al., 1994*). Other studies involved co-immunoprecipitation experiments that revealed the existence of free heavy chain-dimers of different murine MHC I allotypes by pull-down with conformation-specific antibodies. However, it could not be excluded that the detected interactions were enhanced, or indeed caused, by detergents after cell lysis. Additional pulse-chase experiments confirmed that MHC I proteins associate after they have traversed the medial Golgi, but could not localize them to the cell surface (*Capps et al., 1993*).

Our own co-immunoprecipitation experiments confirm these observations for the murine K$^b$ allotype, but they also lack spatial resolution (*Figure 4*). The differential co-immunoprecipitation of surface-biotinylated MHC I proteins finally confirms the presence of protein-protein associations at the cell surface (*Figure 4—figure supplement 1*), but even this method cannot entirely exclude the involvement of intracellular MHC I proteins.

Our two-hybrid assay finally solves the questions of generation and location for MHC I protein-protein interactions. The assay principle has allowed us to establish a system in which we generate defined conformations of MHC I proteins. The results demonstrate that MHC I protein association depends on the generation of free heavy chains, and together with our previous work

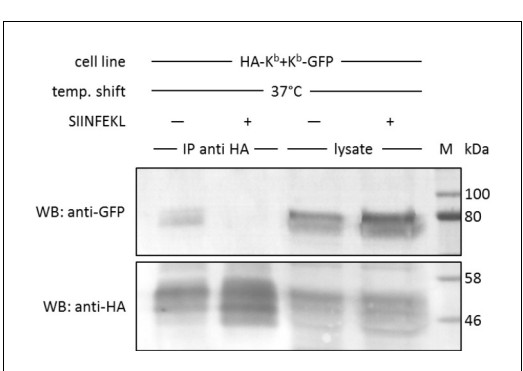

**Figure 4.** *In cis* interactions of K$^b$-GFP and HA-K$^b$ are peptide-dependent and generally occur in cells at 37°C. For co-immunoprecipitation, the same co-transduced cells from the previous experiment were used (STF1/HA-K$^b$ and K$^b$-GFP). Cells were incubated at 25°C overnight to increase K$^b$ cell surface levels and then shifted to 37°C to allow for the dissociation of β$_2$m from the K$^b$ heavy chain. The SIINFEKL (SL8) peptide was added as control to inhibit β$_2$m dissociation (lanes 2 and 4). Cells were then lysed and successfully immunoprecipitated with an anti-HA antibody (bottom row). Immunoisolates and lysate control samples were analysed by western blotting by sequential staining with an anti-GFP antibody (top row) and an anti-HA antibody (bottom row). The K$^b$-GFP construct was specifically co-immunoprecipitated in the absence of peptide, similar to the result on antibody micropatterns (lane 1).
DOI: https://doi.org/10.7554/eLife.34150.008
The following figure supplement is available for figure 4:

**Figure supplement 1.** Co-immunoprecipitation of cell surface K$^b$ molecules with K4-peptide.
DOI: https://doi.org/10.7554/eLife.34150.009

(*Montealegre et al., 2015*), they suggest that these free heavy chains originate from the captured empty $K^bHC/\beta_2m$ dimers at the cell surface upon dissociation of $\beta_2m$. By holding the dimers in the anti-HA patterns and triggering the dissociation of $\beta_2m$ with the 37°C shift, we were able to show that *in cis* interactions are indeed happening at the cell surface (*Figure 3B*). Of course, it is possible that in addition, free heavy chains are generated elsewhere in the cell by $\beta_2m$ dissociation, for example in endosomes, and that they might associate in these locations also.

Is it possible that more than one kind of MHC I protein-protein interaction exists in cells? The MHC I clusters identified by the Yewdell and Edidin groups (*Yewdell, 2006*; *Matko et al., 1994*) contain peptide-bound MHC I proteins. Since our cells were TAP-deficient and thus contained no, or few, peptides for binding to MHC I proteins, we would not have seen the clusters observed by them. If, for example, the Yewdell clusters are formed in the ER then they would not even have formed in our system upon addition of external peptide.

The spatial organization of bait proteins in the plasma membrane allows for the quantification of co-captured proteins into the antibody pattern elements (*Figure 3C*). In our setup, we determined the distribution of prey proteins in control experiments as biological background (this corresponds to our background ratio of 1.1). This background includes also MHC I proteins that are co-captured on the antibody pattern elements before the temperature shift. Whether this background corresponds to pre-formed protein-protein interactions or a heterogeneous surface distribution was not tested and requires detailed analysis.

For MHC I *in cis* interactions at the cell surface, various functional roles have been proposed. They might be a means of accelerated disposal for free heavy chains, preventing re-binding of $\beta_2m$ and peptide and perhaps leading to enhanced internalization and degradation in lysosomes (*Montealegre et al., 2015*). This hypothesis is supported by our finding that associated MHC I proteins do not bind peptide well (*Figure 3—figure supplement 2*) and therefore probably do not interact with TCRs. They might be *in trans* ligands for NK cell receptors or similar proteins, perhaps signaling stress or activation states (*Garcia-Beltran et al., 2016*; *Burian et al., 2016*). We cannot entirely exclude that the associated MHC I proteins contain some $K^bHC/\beta_2m$ dimers that are peptide-receptive, as has been suggested for human MHC I oligomers (*Bodnár et al., 2003*), but free heavy chains are clearly essential for *in cis* interactions, since single-chain $K^bHC/\beta_2m$ dimers do not associate (*Figure 3A*). Another possibility is that the associated free heavy chain might influence the surface levels of other proteins with which MHC I proteins are known to interact, such as APLP or insulin receptor, thereby mediating non-immunological functions of MHC I proteins (*Tuli et al., 2008*; *Shatz, 2009*; *Dixon-Salazar et al., 2014*). Our assay is a promising tool to extend the interaction studies for MHC I proteins by the proposed interaction partners.

We have shown here the formation of homotypic *in cis* interactions of the murine MHC I allotype $K^b$. Interestingly, previous work suggests that the tendency of *in cis* interactions varies among MHC I allotypes (*Capps et al., 1993*). Thus, it was hypothesized that those MHC I allotypes that do not associate are not internalized (by the accelerated disposal mechanism proposed above) and that they will bind exogenous peptides to provoke autoimmune reactions (*Capps et al., 1993*). This might be interesting in the case of those subtypes of HLA-B27 that are implicated in inflammatory autoimmune diseases such as spondyloarthropathies (*Chen et al., 2017*; *Allen et al., 1999*). For HLA-B*27:05, formation of heavy chain dimers at the cell surface, or in early endocytic compartments, was demonstrated to occur through a disulfide bond between the unpaired cysteine-67 residues. Since $K^b$ does not have an unpaired cysteine in the extracellular domain, this type of dimerization is not possible for $K^b$. Still, the interesting possibility exists that the *in cis* heavy chain associations described by us for $K^b$ might also occur with B*27:05 and might cause the formation of the covalent B*27:05 dimers. We look forward to future investigations.

Due its versatility, our assay allows for the development of a screen to test for the tendency of individual MHC I allotypes to associate with themselves, and with other allotypes. This may be extended to human MHC I proteins, whose empty dimers can be enriched at the cell surface by incubation with low-affinity dipeptide ligands (*Saini et al., 2015*), and even to the empty forms of HLA-F that were recently discovered to bind NK cell receptors (*Garcia-Beltran et al., 2016*; *Burian et al., 2016*). Consequently, by its application to the human system, this screening tool can be developed to investigate the correlation between cell surface protein-protein interactions and human autoimmune disease. Generation of anti-HA antibody micropatterns by microcontact

printing on conventional glass coverslips makes this assay especially suitable for such high-through-put approaches.

In addition to its demonstrated application in the detection of conformation-dependent *in cis* interactions, the assay can be further developed towards more detailed analysis. One possibility is the integration of conventional immunostaining for the identification of other proteins involved in the redistribution of co-captured proteins. For MHC I proteins, for example, observing the accumulation of adaptor proteins involved in endocytic processes (e.g. Rab proteins) on the pattern elements under condition of co-capture will contribute to understand the nature of MHCI protein endocytosis and the functional role of *in cis* interactions.

Another possible technical development is to combine the assay with fluorescence revovery after photobleaching (FRAP) measurements to test the dynamics of the interactions (i.e. dissociation and re-association). Such a combined two-hybrid-FRAP assay is potentially superior to conventional (FRET) experiments, since the enrichment of proteins in the pattern elements increases the abundance of the interaction partners and might thus enable the detection of very weak interactions.

# Materials and methods

## Key resources table

| Reagent type (species) or resource | Designation | Source or reference | Identifiers | Additional information |
|---|---|---|---|---|
| Gene (mus musculus) | H-2K$^b$ | NCBI GenBank | NM_001001892.2 | |
| Gene (mus musculus) | H-2D$^b$ | NCBI GenBank | NM_010380.3 | |
| Cell line (homo sapiens) | STF1 | PMID:10074495 | N/A | |
| Transfected construct (mus musculus) | STF1/K$^b$-GFP | This paper | N/A | Stable cell lines were generated by lentiviral transduction and antibiotic selection as described in PMID: 24806963. |
| Transfected construct (mus musculus) | STF1/D$^b$-GFP | This paper | N/A | |
| Transfected construct (mus musculus) | STF1/E3 HA-K$^b$-GFP | This paper | N/A | |
| Transfected construct (mus musculus) | STF1/E3 HA-K$^b$ +K$^b$-GFP | This paper | N/A | |
| Transfected construct (mus musculus) | STF1/E3 HA-K$^b$ +K$^b$-h$\beta_2$m-GFP (single chain) | This paper | N/A | |
| Antibody | Y3 | PMID: 6181513 | N/A | Produced and purified in house from hybridoma cells. 0.6 µg/µL for printing |
| Antibody | 27-11-13S | PMID: 6935293 | N/A | |
| Antibody | 12CA5 (Hemagglutinin; HA) | PMID: 6192445 | N/A | Produced and purified in house from hybridoma cells 0.6 µg/µL for printing; 1:100 dilution of hybridoma supernatant for Western blotting. |
| Antibody | BBM.1 | PMID: 91522 | N/A | Produced and purified in house from hybridoma. 0.1 µg/µL for staining. |
| Antibody | rabbit anti-GFP | Abcam | Cat #: ab290, RRID:AB_303395 | 1:1000 |
| Antibody | rabbit antisera against H-2K$^b$ and H-2D$^b$ | Charles River Laboratories | Rabbits were immunized with a peptide corresponding to residues 331–349 of the cytoplasmic tail of both heavy chains | 1:1000 |
| Antibody | goat anti-rabbit IgG-AP | Bio-Rad | Cat #: 170–6518, RRID:AB_11125338 | 1:10000 |
| Antibody | donkey anti-mouse IgG Alexa Fluor 568 | Thermo Fisher Scientific | Cat #: A10037, RRID:AB_2534013 | 1:400 for staining antibody micropatterns |

*Continued on next page*

*Continued*

| Reagent type (species) or resource | Designation | Source or reference | Identifiers | Additional information |
|---|---|---|---|---|
| Recombinant DNA reagent | puc2CL6Ipwo (lentiviral vector) | PMID: 21248040 | N/A | |
| Recombinant DNA reagent | puc2CL6IPwo/ K$^b$-GFP (plasmid) | This paper | N/A | C-terminally GFP-tagged H-2K$^b$ and H-2D$^b$ cDNA were cloned into the lentiviral vector via the *Xho*I and *Age*I sites. |
| Recombinant DNA reagent | puc2CL6IPwo/ D$^b$-GFP (plasmid) | This paper | N/A | |
| Recombinant DNA reagent | puc2CL6IPwo/E3 -HA-K$^b$-GFP (plasmid) | This paper | N/A | N-terminally E3-tagged and HA-tagged) H-2K$^b$ (±GFP) cDNA were cloned into the lentiviral vector via the *Xho*I and *Age*I sites. |
| Recombinant DNA reagent | puc2CL6IPwo/ E3-HA-K$^b$ (plasmid) | This paper | N/A | |
| Recombinant DNA reagent | puc2CL6IPwo/ E3-HA-K$^b$ (plasmid) | This paper | N/A | |
| Recombinant DNA reagent | puc2CL6IPwo/K$^b$- hβ$_2$m-GFP (plasmid) | This paper | N/A | Cloning strategy of the single chain construct according to PMID: 16049493 |
| Sequence-based reagent | forward primer E3 tag | This paper | N/A | 5´-ACTCAGACCCGCGCGGGCGAGATCG CAGCTCTGGAGAAGGAGATTGCCGCAT TGGAGAAGGAGATAGCGGCACTCGAG AAGTATCCATACGACGTCCC-3´ |
| Sequence-based reagent | reverse primer E3 tag | This paper | N/A | 5´-GGGACGTCGTATGGATACTTCTCGA GTGCCGCTATCTCCTTCTCCAATGCGG CAATCTCCTTCTCCAGAGCTGCGAT CTCGCCCGCGCGGGTCTGAGT-3´ |
| Sequence-based reagent | forward primer HA tag (1/2) | This paper | N/A | 5´-CCGACTCAGACCCGCGCGGGCC CATATCCATACGACGTCCCACACTC GCTGAGGTATTTCGTCACC-3´ |
| Sequence-based reagent | forward primer HA tag (1/2) | This paper | N/A | 5´-GGCCCATATCCATACGACGTCCCAG ATTATGCCGGCGGTGGACACTCGCTG AGGTATTTCGTCACC-3´ |
| Sequence-based reagent | reverse primer HA tag | This paper | N/A | 5´-GGTGACGAAATACCTCAGCGAGTG TGGGACGTCGTATGGATATGGGCCCG CGCGGGTCTGAGTCGG-3´ |
| Sequence-based reagent | reverse primer HA tag | This paper | N/A | 5´-GGTGACGAAATACCTCAGCGAGTG TCCACCGCCGGCATAATCTGGGACGTCG TATGGATATGGGCC-3´ |
| Peptide, recombinant protein | SIINFEKL peptide | GeneCust Ellange, Luxemburg | N/A | 2 µM final concentration |
| Peptide, recombinant protein | SIINFEKL$^{TAMRA}$ peptide | GeneCust Ellange, Luxemburg | N/A | 2 µM final concentration |
| Peptide, recombinant protein | K4$^{Atto\ 633}$ peptide | emc microcollections | N/A | 25 nM final concentration |
| Peptide, recombinant protein | K4$^{bio}$ peptide | emc microcollections | N/A | 200 nM final concentration |
| Chemical compound, drug | Alexa Fluor 647 NHS ester | Thermo Fisher Scientific | Cat #: A37566 | |
| Chemical compound, drug | Atto 542 NHS ester | ATTO-TEC, | Cat #: AD 542–31 | |
| Software, algorithm | ImageJ | National Institutes of Health | | |

## Photolithography

Silicon master molds were prepared by semiconductor photolithography as described previously. See *Dirscherl et al. (2017)* for details.

## PDMS stamps and antibody patterns

PDMS stamps were generated from basic elastomer and curing agent (Sylgard 184 Silicone Elastomer Kit) from Dow Corning (Midland, USA) mixed in a 10:1 ratio. The prepared stamps were inked

with the indicated antibody solutions and then placed on round microscopy glass coverslips (#1, 22 mm). See *Dirscherl et al. (2017)* for details.

## Patterning cell surface proteins

Coverslips were placed into sterile 6-well plates. Cells were immediately seeded as indicated at a concentration of ≈50.000 cells per well and incubated on the antibody patterns. Usually, cells were incubated for 4–6 hr at 37°C for adhesion and then shifted to 25°C overnight to accumulate $K^b$ molecules at the cell surface for a better signal-to-noise ratio of patterned $K^b$ molecules. For co-capturing experiments, samples were then kept at 25° C or shifted back to 37°C for 3–4 hr to allow for the dissociation of $\beta_2 m$.

For each experiment fresh antibody micropatterns are generated, representing technical replicates. The experiments were individually repeated on different days at least three times and representative images are shown in the respective figures. For each experimental condition, at least triplicates were performed per experiment. In this setup, the individual cells seeded onto the antibody micropatterns represent the biological replicates.

## Antibodies

Mouse monoclonal hybridoma supernatants Y3 (against the complex of $K^b$ free heavy chain with $\beta_2 m$ (*Hämmerling et al., 1982*), 27-11-13S (against the complex of $D^b$ free heavy chain with $\beta_2 m$ (*Ozato and Sachs, 1981*), hemagglutinin (HA) 12CA5 (*Niman et al., 1983*), and BBM.1 (*Brodsky et al., 1979*) were described previously. Antibodies for immunoprecipitation were rabbit anti-GFP (Abcam ab290), rabbit antisera against H-2$K^b$ and H-2$D^b$ (Charles River Laboratories, Kisslegg, Germany), and goat anti-rabbit IgG-AP conjugate (1706518, Biorad, Munich, Germany). Secondary antibody against the HA-antibody was donkey anti-mouse IgG Alexa Fluor 568 (a10037, Thermo Fisher Scientific, Darmstadt, Germany).

## Dyes

Purified antibodies were either labeled with the Alexa Fluor−647 NHS ester (Y3, 27-11-13S and 12CA5) or with the Atto 542 NHS ester (BBM.1) according to the manufacturers´ protocols. Alexa Fluor−647 NHS was obtained from Thermo Fisher Scientific (Darmstadt, Germany) and the Atto 542 NHS from ATTO-TEC (Siegen, Germany).

## Peptides

Peptides were synthesized by GeneCust (Ellange, Luxemburg) and emc microcollections (Tübingen, Germany) and purified by HPLC (90% purity). The $K^b$-specific peptide SL8 (SIINFEKL in the single-letter amino acid code) was labeled with the fluorescent dye 5'-carboxytetramethylrhodamine (TAMRA) on the lysine side chain (avoiding interference with peptide binding to $K^b$) to give SIINFEKL$^{TAMRA}$. Labeled and unlabeled peptides were added to the cells at a final concentration of 2 μM for 15–30 min at 37°C to detect peptide binding. Cells were then washed with phosphate buffered saline (PBS, 10 mM phosphate pH 7.5, 150 mM NaCl), fixed, and observed by confocal laser scanning microscopy (cLSM).

## Cell lines and gene expression

For experiments, we used STF1 TAP-deficient human fibroblasts (kindly provided by Henri de la Salle, Etablissement de Transfusion Sanguine de Strasbourg, Strasbourg, France; see (*de la Salle et al., 1999*) for reference). These cells were chosen to make sure that only the murine allotype H-2$K^b$ was investigated, without interference of other murine MHC I allotypes that might cross-react with the used antibodies. STF1 cells were authenticated by HLA haplotype genotyping; they were regularly tested for mycoplasma. The cells were stably transduced with $K^b$-GFP, $D^b$-GFP, and E3-HA-$K^b$-GFP. Lentiviruses were produced and used for gene delivery as described previously (*Hein et al., 2014*). For co-capturing experiments, co-transduced STF1 cells were used. The cells were first selected with puromycin for E3-HA-$K^b$ (the additional E3 tag (EIAALEK)$_3$ is a 21 amino-acid long extracellular tag that was initially introduced for co-staining experiments). Cells were then transduced with the indicated $K^b$-GFP constructs and h$\beta_2 m$ where indicated and again selected with

puromycin to obtain STF1/E3-HA-K$^b$+K$^b$ GFP or STF1/E3-HA-K$^b$+K$^b$-hβ2m-GFP (single chain K$^b$, single chain construct in which the light chain β$_2$m is fused by a linker to the K$^b$ heavy chain).

## Staining with the K4 peptide

The K4 peptide was synthesized by emc microcollections (Tübingen, Germany). The E3-tag specific peptide K4 (*Litowski and Hodges, 2002*; *Yano et al., 2008*) (a 28 amino-acid long peptide abbreviated as (KIAALKE)$_4$ in the single-letter amino acid code) was labeled with an Atto 633 fluorophore at the N terminus. For co-staining experiments, cells were fixed with 3% paraformaldehyde (PFA), washed, and permeabilized with 0.1% Triton X-100. The K4$^{Atto633}$ peptide was added to the cells at a final concentration of 25 nM in PBS and incubated for 5 min at RT to stain the transduced E3-HA-K$^b$ construct.

## Immunofluorescence stainings

For antibody co-staining experiments, cells were fixed with 3% PFA, washed and permeabilized with 0.1% Triton X-100 and stained with the respective antibodies.

## Microscopy

A confocal laser scanning microscope (LSM 510 Meta, Carl Zeiss Jena GmbH, Germany) equipped with argon and helium-neon lasers at 488, 543 and 633 nm was used. Images were recorded with a 63x Plan Apochromat oil objective (numerical aperture 1.4) at a resolution of 1596 × 1596 pixels. Data acquisition was performed with the LSM 510 META software, release 3.2 (Carl Zeiss Jena GmbH). During image acquisition, patterns and cells were imaged in the same focal plane at a pinhole of ≈1 Airy unit. Image analysis and processing were performed using ImageJ (National Institutes of Health, Bethesda, USA). Image processing comprises cropping, rotation and adjustment of brightness and contrast levels.

Generally, all cells on the antibody micropatterns were scanned by eye, and representative cells were chosen for image acquisition. Due to the range of expression levels, only cells with a moderate expression level were selected for evaluation. Only adhered cells were evaluated; cells with altered morphologies (e.g. apoptotic cells) were excluded.

## Quantification of co-captured proteins

To compare the spatial distribution of K$^b$-GFP between the pattern elements and the pattern element interspaces, we determined the mean fluorescence intensity of GFP in the entire cell area and also for the areas on the pattern elements (ImageJ, National Institutes of Health, Bethesda, USA). The ratio of fluorescence intensity on pattern elements over total fluorescence intensity was calculated. Control experiments (peptide addition, incubation at 25°C and the single chain construct) gave a 1.1 ratio, and thus this was defined as the background signal. The co-capturing experiments usually showed a ratio of 1.3 (*Figure 3C*). For each condition, ≥ 14 cells of ≥ 2 individual experiments were used.

## Co-immunoprecipitation

For co-immunoprecipitation with the anti-HA antibody, co-transduced (E3-HA-K$^b$+K$^b$-GFP) and selected STF1 cells were incubated at 25°C overnight. The next day, cells were incubated in presence or absence of 10 µM SL8 for 10 min at 25°C, then shifted to 37°C for 15 min, trypsinised and harvested. Cell pellets were lysed in native lysis buffer (50 mM Tris-Cl (pH 7.4), 150 mM NaCl, 5 mM EDTA, and 1% Triton X-100) for 1 hr at 4°C. After lysis, the supernatant was immunoprecipitated with the anti-HA antibody for 30 min at 4°C. Beads were washed and resuspended in Laemmli sample buffer (LSB) buffer and boiled at 95°C for 10 min. The immunoisolates were separated by SDS-PAGE and immunoblotted sequentially with an anti-GFP antibody and an anti-HA antibody. The experiment was performed three times.

For co-immunoprecipitation of the cell surface K$^b$ molecules with the biotinylated K4 peptide (K4$^{biotin}$), co-transduced (E3-HA-K$^b$+K$^b$-GFP) and selected STF1 cells were incubated at 25°C overnight. Next day, cells were incubated in presence or absence of 10 µM SL8 for 10 min at 25°C. E3-tagged K$^b$-molecules were labeled with K4$^{biotin}$ for 5 min at room temperature using 200 nM biotinylated K4 peptide. Following biotinylation, cells were placed at 37°C for 15 min to allow for co-

capture. Cells were collected into native lysis buffer (50 mM Tris-Cl (pH 7.4), 150 mM NaCl, 5 mM EDTA, and 1% Triton X-100) by scraping and lysed for 45 min at 4°C. Biotinylated E3-HA-$K^b$ (with $K4^{biotin}$) was immunoprecipitated from post-nuclear supernatants using neutravidin-agarose beads (Thermo Fisher Scientific, Darmstadt Germany). Beads were washed in lysis buffer and wash buffer (50 mM Tris-Cl (pH 7.4), 150 mM NaCl, 5 mM EDTA, and 0.1% Triton X-100) and supplemented with endogylcosidase F1 for 2 hr at 37°C, or left untreated. Isolated proteins were retrieved from beads by boiling and separated by SDS-PAGE. $K^b$-GFP was detected by anti-GFP antiserum and E3-HA-$K^b$ was detected by 12CA5 (anti-HA) following western blotting. The experiment was performed twice.

## Acknowledgements

The authors thank Susanne Illenberger und Susanne Fritzsche for suggestions on the manuscript; and Ursula Wellbrock for the cultivation of hybridoma cell lines and excellent technical assistance.

## Additional information

### Funding

| Funder | Grant reference number | Author |
| --- | --- | --- |
| Deutsche Forschungsgemeinschaft | SP583/7-2 | Sebastian Springer |
| Tönjes Vagt Foundation | XXXII | Sebastian Springer |
| Bundesministerium für Bildung und Forschung | 031A153A | Sebastian Springer |

The funders had no role in study design, data collection and interpretation, or the decision to submit the work for publication.

### Author contributions

Cindy Dirscherl, Conceptualization, Supervision, Validation, Investigation, Visualization, Methodology, Writing—original draft; Zeynep Hein, Conceptualization, Investigation, Writing—review and editing; Venkat Raman Ramnarayan, Catherine Jacob-Dolan, Investigation, Acquisition of data, Analysis, Interpretation of data; Sebastian Springer, Conceptualization, Supervision, Funding acquisition, Project administration, Writing—review and editing

### Author ORCIDs

Cindy Dirscherl http://orcid.org/0000-0002-8973-2835
Zeynep Hein http://orcid.org/0000-0002-6335-8961
Sebastian Springer http://orcid.org/0000-0002-5527-6149

### Decision letter and Author response

Decision letter https://doi.org/10.7554/eLife.34150.013
Author response https://doi.org/10.7554/eLife.34150.014

## Additional files

### Supplementary files

• Transparent reporting form
DOI: https://doi.org/10.7554/eLife.34150.010

### Data availability

All data generated or analysed during this study are included in the manuscript and supporting files. Source data files have been provided for Figure 3C.

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
