## [Decision Letter]

Thank you for submitting your article "A novel two-hybrid antibody micropattern assay reveals conformation-specific cell surface clustering of MHC I proteins" for consideration by *eLife*. Your article has been reviewed by three peer reviewers, including Michael L Dustin as the Reviewing Editor and Reviewer #1, and the evaluation has been overseen by Arup Chakraborty as the Senior Editor.

The reviewers have discussed the reviews with one another and the Reviewing Editor has drafted this decision to help you prepare a revised submission.

Summary:

The reviewers agreed that the use of micro contact printing to evaluate lateral interactions MHC class I proteins in situ was novel and presented a useful new approach to such problems. There was a consensus that the tool would be more powerful if the patterns were quantified and it was felt that this would be straight forward as the patterns are user defined and should be easy to segment and measure. This might allow application to weaker interactions where results may not as obvious.

Essential revisions:

1) Quantify imaging results and validate for this example. Given this signal to noise, how weak an association could be detected?

2) We would strongly recommend not using the word cluster to describe the patterning of class I molecules with glass bound Abs. While this is clearly a useful technique, it is highly artificial, and for all we know, each "cluster" could be a single pair of co-localized Abs. Cluster has been used (appropriately) in the past to refer to spontaneously concentrated class I molecules, likely present in membrane domains that limit their ability to diffuse from each other. The use of cluster in the present study muddies the waters considerably, and will likely increase confusion, rather than the opposite.

3) Is it possible to reverse the association of non-conformed K^b^ molecules by addition peptide and or β_2_m?

4) Subsection “Stabilizing effect of conformation-specific antibodies allows for differential patterning of K^b^ dimers and free heavy chains”, first paragraph: regarding the use of TAP negative cells (whose original publication should be cited), Day et al., 1994, reported that TAP positive cells actually express more peptide receptive cell surface molecules, undermining the reasoning of these statements.

5) "So their endogenous class I molecules would not interfere with the Ab micropatterns". This should be rephrased as it is not clear.

[Editors' note: further revisions were requested prior to acceptance, as described below.]

Thank you for submitting your article "A novel two-hybrid antibody micropattern assay reveals conformation-specific cell surface clustering of MHC I proteins" for consideration by *eLife*. Your article has been reviewed by two peer reviewers, and the evaluation has been overseen by a Reviewing Editor and Arup Chakraborty as the Senior Editor. The reviewers have opted to remain anonymous.

Summary:

The original reviewers agree that you have provided a thoughtful response to essential revisions proposed, but still raise concerns about the current version that prevent it from being published in *eLife*. The reviewing editor has now carefully examined these remaining concerns including a review of the literature regarding in situ analysis of protein interactions using microcontact printing.

Essential revisions:

1) The current application of the method is thought to be of high value and provides significant biological insight. The method itself is too conceptually similar to earlier work from the Schütz lab (which you have cited correctly) to be referred to as a "novel" assay and thus it is requested that you remove the word "novel" from the title and anywhere else in the manuscript this is claimed, unless you specifically identify the novel aspect. The reviewers acknowledge that you have literally created a "two-hybrid" version based on using the epitope tagged bait (hybrid 1) and the fluorescent tagged prey (hybrid 2), which Schütz didn't fully realise, but they had already used the bait-prey terminology in their 2010 Jove paper, and conceptually it is too small an advance in terms of assay technology. They didn't need to make a first hybrid as they tested a transmembrane bait interacting with a cytoplasmic prey. But they have everything else you describe and more sophisticated analysis.

2) While the reviewers appreciate that you define cluster in the context of this study, "cluster" with respect to MHC class I antigens has been previously defined and this needs to be respected within papers in this field to avoid confusion. It is also felt that with the ability to detect only a 10% steady state association with the current method (based on use of 1.1 as a threshold) that you apparently can't detect the earlier defined MHC class I clusters. But you are not saying these don't exist. A compromise would be to refer to the phenomenon you are studying as "induced clusters" throughout to distinguish them from previously described spontaneous clusters of MHC class I trimers, that you neglect due to your detection threshold. It is essential to clarify this or use a term not including "cluster".

3) Proposed new title: "A two-hybrid antibody micropattern assay reveals conformation-specific cell surface induced clustering of MHC I proteins".

4) You describe the quantitative threshold of 1.1 in the response to reviewers, but we couldn't find this in the paper. Please include in the Materials and methods the 1.1 threshold for defining an induced cluster in the system. It is not clear how the gap between 1.1 and 1.3 is defined as a grey zone in the rebuttal, but you should also mention this if you feel it is important to actually set the threshold at 1.3, although this is very close to your experimental result.

[Editors' note: further revisions were requested prior to acceptance, as described below.]

Thank you for choosing to send your work entitled "A novel two-hybrid antibody micropattern assay reveals cell surface clustering of MHC I heavy chains" for consideration at *eLife*. Your article has been reviewed by a Senior Editor and a Reviewing editor, and we are prepared to consider a revised submission with no guarantees of acceptance.

There is no editorial policy against use of "novel" in titles of *eLife* papers, but this has been infrequent – at around 0.2%. In the first decision letter the editors did override one reviewer in acknowledging that the approach was novel, but when the reviewer persisted in questioning this an editor went to the primary literature and looked at the precedents specifically to address this issue independently. As described in the second decision letter the opinion of the editors was then changed to agree with the reviewer that the core of the assay technology was well described earlier and that this assay is a novel application of an existing assay concept. Since this is too much information to convey in the title, the decision was to remove "novel" from the title and then explain the more nuanced novelty of this application in the text. The editors still ask that you remove novel from the title.

The second issue has to do with the definition of cluster and whether this is misleading in this context. The editors still ask that you change this terminology to avoid confusion and quantitative implications of the term. The interaction leading to a signal in this assay could be sub-stoichiometric, 1:1 or super-stiochiometric. The term cluster suggests groupings of greater than 1:1, because 1:1 would be called a dimer, and less than 1:1 would not be cluster. This method doesn't provide any information about stoichiometry, but just that it leads to co-localization of prey with the bait by fluorescence microscopy. The heavy chain interaction you are detecting may very well be a dimer, based on other literature, but this cannot be determined from this data. So describing the process as a cluster seems to be an over-interpretation. The bait and pray are simply co-localized at conventional visible light resolution. The authors should refer to co-localization, co-capture or enrichment to precisely describe what the assay detects as a ratio of on/off pattern as quantified in Figure 3?

The last issue was the baseline co-localization of prey with bait, which is described as generating a ratio of 1.1 of on/off pattern. This ratio was 1.3 for conditions leading to the empty MHC I heavy chains. Is 1.1 measured under conditions favoring the full heterotrimer of MHC I, peptide and β_2_m, significantly different than 1.0? The quantification was new data added by the authors to Figure 3. This is where 10% came from. The authors may not have data to determine if 1.1 is significantly different than 1.0, but as a discussion point, they might discuss that the previously describe interactions of MHC I proteins on cells may be captured in this number, which would then be a biological background for this assay method and thus will impact its sensitivity. This would also put the relative magnitude of other forms of MHC clustering and this phenomenon in context. More experiments would be needed to sort this out- probably use of different controls to find proteins that are totally unaffected by capture of some of the MHC I molecules to the patterned antibodies. The authors are urged to put these aspects of this assay and earlier investigations of MHC I distribution on the cell surface in context as best they can.

---

## [Author Response]

Essential revisions:1) Quantify imaging results and validate for this example. Given this signal to noise, how weak an association could be detected?

We thank the reviewer for this important comment. In our revised manuscript, we have now included the quantification of K^b^-K^b^ cluster formation at 37 °C and the respective controls according to our observations in Figure 3B. We have thus added the quantification to Figure 3 (see new Figure 3C) and have extended the Results section (subsection “Antibody micropatterns reveal conformation-dependent *in cis* interactions of K^b^ free heavy chains”, fifth paragraph) as well as the Materials and methods section accordingly (subsection “Quantification of clustering”).

Briefly, for the quantification we used ImageJ to compare the mean fluorescence intensity of the entire cell with the mean fluorescence intensity of K^b^-GFP in the areas of the pattern elements. A relocalization of K^b^-GFP to the pattern elements leads to increased fluorescence intensity levels on the pattern elements and an increase of the ratio of the fluorescence intensity of the pattern elements over the fluorescence intensity of the entire cell. According to theoretical considerations and the observations in our system, we interpret a ratio of 1.3 as clustering and values below 1.1 as background signal (Figure 3C). Quantification is not trivial in our example since the K^b^-transfected TAP-deficient STF1 cells have a generally strong ER background. This is because many K^b^ molecules that cannot be loaded with intracellular peptides are not transported to the plasma membrane. Despite these high background levels, we obtained significant differences between clustering cells and controls (see Figure 3C).

We think that the numerical fluorescence ratios might be different for other protein-protein interactions, depending on cell type, expression levels, background signal, and also the pattern element sizes. These factors will play important roles for the sensitivity of the assay but can be adjusted during optimization experiments.

Regarding the detection of weak protein-protein interactions asked by the reviewer, we think that our method can be even advantageous over conventional methods, since it allows for the accumulation of proteins to the pattern elements over time to increase the signal intensity. Also, one can increase the number of captured proteins by increasing the antibody concentration or achieve better contrast by varying pattern geometries and/or pattern element sizes. Alternatively, a readout by TIRF or super-resolution microscopy techniques might allow researchers to detect weak signals when only shorter incubation times on the antibody micropatterns can be realized, or when protein-protein interactions require temporal resolution.

2) We would strongly recommend not using the word cluster to describe the patterning of class I molecules with glass bound Abs. While this is clearly a useful technique, it is highly artificial, and for all we know, each "cluster" could be a single pair of co-localized Abs. Cluster has been used (appropriately) in the past to refer to spontaneously concentrated class I molecules, likely present in membrane domains that limit their ability to diffuse from each other. The use of cluster in the present study muddies the waters considerably, and will likely increase confusion, rather than the opposite.

We thank the reviewer for this comment, which has revealed to us an important potential misunderstanding. To understand our assay, our readers need to be able to discriminate between two events that we have termed captureand clustering. We still believe that this terminology is precise and useful according to the following definitions:

1) We use the term capture to describe the binding of cell surface membrane proteins to the antibody micropatterns. Capture thus describes the direct antibody-antigen interaction that prevents internalization of the bound proteins, i.e., the mechanical trapping of K^b^ proteins at the cell surface (see Figure 3).

2) In contrast, we use the term clustering to describe the binding of further protein molecules to the already captured proteins on the pattern elements. This is the event that our assay is set up to detect. This definition matches the statement of the reviewer: “Cluster has been used (appropriately) in the past to refer to spontaneously concentrated class I molecules, likely present in membrane domains that limit their ability to diffuse from each other.”

It is essential that the two terms are clearly introduced, differentiated, and understood by the readers. We very much agree that the term *clustering* might be misleading if not properly defined, but we are convinced that *clustering* is still the best term for our observation. We have therefore introduced a clearer definition of both terms in the manuscript (Introduction, second and third paragraphs). In addition, we have gone through our Introduction to make sure that the term *capture* is now stringently used.

With these adjustments, we now trust that our explanations are clear and straightforward, and that the readers will be able to distinguish between the two terms.

3) Is it possible to reverse the association of non-conformed K^b^ molecules by addition peptide and or β_2_m?

We thank the reviewer for pointing out this exciting possibility. Indeed, during our work, we have thought hard about the molecular structure of the clustered proteins, and especially if they would be able to bind peptide and/or β_2_m.

First, we tested peptide binding to the clustered K^b^ molecules in order to characterize the clustered K^b^ molecules in more detail (see Figure 3—figure supplement 2). In these experiments, we observed that peptide binding was impaired, suggesting that the clustered K^b^ molecules are no longer peptide-receptive. This is plausible since MHC class I free heavy chains do not usually bind peptide well.

Second, with respect to β_2_m binding, we have – in previous published work – performed β_2_m incubation experiments in the same experimental system of K^b^ in STF1 cells and found that exogenous β_2_m does not significantly increase the lifetime of surface class I, suggesting that it will not significantly re-bind to the K^b^ free heavy chain once the original β_2_m has dissociated (Montealegre et al., 2015).

In the work presented in this manuscript, we show that incubation with peptide prevents the formation of clusters (Figure 3B, column 4). We have not incubated the cells with peptides or with exogenous β_2_m after cluster formation in order to try to reverse the K^b^-K^b^ interaction, as the reviewer suggested, but from the above observations of impaired peptide and β_2_m binding, we assume that K^b^ free heavy chain clusters will not be dispersed by the addition of β_2_m or peptide.

We believe that the detailed molecular mechanism of cluster formation, its potential dynamic nature, and the subsequent fate of the class I free heavy chain clusters are very interesting questions to follow up on, but we would like to leave these investigations to a future manuscript.

4) Subsection “Stabilizing effect of conformation-specific antibodies allows for differential patterning of K^b^ dimers and free heavy chains”, first paragraph: regarding the use of TAP negative cells (whose original publication should be cited), Day et al., 1994, reported that TAP positive cells actually express more peptide receptive cell surface molecules, undermining the reasoning of these statements.

We thank the reviewer for carefully reading our manuscript. In our revised manuscript we have now added the original publication of STF1 cells (de la Salle et al., 1999) specifically to the mentioned sections as requested. We have also cited the Day et al., 1995 paper.

As the reviewer commented correctly, Day and collaborators investigated the role of TAP for the generation of peptide-receptive molecules at the cell surface. In their work, they found that the murine TAP-positive RMA cells have higher surface levels of peptide-receptive K^b^ molecules than TAP-deficient RMA-S cells (Day et al., 1995).

For our studies, we opted for a human TAP2-deficient cell line and transduced it with the K^b^ constructs described in the manuscript in order to bypass any murine MHC class I background issues in our system. Prompted by the question of the reviewer, we became curious as to the levels of peptide-receptive K^b^ molecules on the surface of the STF1 cells, and similar to the work of Day et al., we have now compared STF1 cells with STF1+TAP2 cells (STF1 cells transduced with TAP2, which have wild type TAP function). We transduced both cell lines with HA-K^b^, thus generating the two stable cell lines (STF1/HA-K^b^ and STF1+TAP2/HA-K^b^). To quantify peptide-receptive molecules, we added the K^b^-specific high-affinity peptide SIINFEKL (SL8; 10 µM) to the cell culture medium and incubated the cells for 20 min at 37 °C. The cells were then washed, trypsinized, and stained with antibodies according to the standard protocol for flow cytometry.

**Author response image 1. respfig1:** Comparison of cell surface expression of HA-K^b^ in STF1 and STF1+TAP2 cells by flow cytometry. STF1 cells were transduced with HA-K^b^ and stained with anti-HA or 25-D1.16 and anti-mouse IgG conjugated to Alexa Fluor 488 and subjected to flow cytometry. (**A**) Surface intensities of HA-K^b^ of both cell lines are represented as bar charts. TAP-deficient STF1 cells are represented in black and TAP2-proficient cells (STF1+TAP2) are represented in grey. Cells were incubated with (+) and without (-) 10 µM of the high affinity peptide SL8 and stained with the indicated antibodies. (**B**) The increase in surface signal after peptide addition in (**A**) for both antibodies is displayed as ratio. (n= 3; standard deviations as indicated).

The total K^b^ surface levels were determined by staining with the anti-HA antibody (Author response image 1). In STF1 cells, K^b^ surface levels only showed a small and barely significant increase upon addition of SL8, probably because the peptide prevented endocytosis of some empty dimers (Montealegre et al., 2015). The K^b^ surface levels in STF1+TAP2 cells remained the same upon peptide addition, although they have generally higher K^b^ surface levels (Author response image 1).

In order to determine the peptide-receptive population of the detected K^b^ molecules, the same samples were stained with the 25-D1.16 antibody, specifically binding to K^b^ molecules that are loaded with the SL8 peptide (Porgador et al., 1997). As expected, the 25-D1.16 signal increases clearly upon addition of the SL8 peptide (see Author response image 1).

Remarkably, the almost identical increase of the 25-D1.16 signal after peptide addition shows that the levels of peptide-receptive molecules are similar in both cell lines. This suggests that in both cell lines, despite their very different K^b^ surface levels (Author response image 1), similar mechanisms are at work to limit the amount of peptide-receptive ('empty') class I at the cell surface.

These findings are in contrast to the findings of Day and collaborators on RMA-S cells that the reviewer mentions. We are not surprised, though, since class I quality control and trafficking parameters vary between different cells; one important difference is that RMA-S cells are lymphocytes, in which the antigen processing and presentation genes in the MHC are fully induced (whereas STF1 cells are fibroblasts).

Our original reason for using TAP-deficient STF1 cells have to do with the homogeneity of the class I population. The experiments in our manuscript rely on our ability to generate free heavy chains of H-2K^b^ on the cell surface, and to trap them there. This is achieved by capturing the K^b^ heavy chain/β_2_m dimers and incubating the cells at 37 °C to allow these dimers to dissociate, leaving captured free heavy chain that is unable to bind peptide (see answer to comment 3 above, and Figure 2B, column 3, of the manuscript). These free heavy chains then cluster with other free heavy chains (Figure 3B, column 3). Since we use TAP-deficient STF1 cells, the majority of the K^b^ molecules that reach the surface are probably heavy chain/β_2_m dimers, which provides us with a relatively homogeneous population. If, according to the question of the reviewer, we were to use STF1+TAP2 cells, then we would additionally have a large population of K^b^ heavy chain/β_2_m/peptide trimers (compare the STF1 and the STF1+TAP2 columns in Author response image 1), which do not turn into free heavy chains at 37 °C and thus provide a second species of K^b^ at the cell surface that does not undergo clustering and possibly dilutes our readout signal. This, for us, was the main reason for using STF1 cells for the experiments.

We have not made any changes to the manuscript as a result of the work described in this answer.

5) "So their endogenous class I molecules would not interfere with the Ab micropatterns". This should be rephrased as it is not clear.

We thank the reviewer for carefully reading our manuscript and for the suggested corrections. We have changed the indicated sentences in the revised manuscript accordingly (see subsection “Cell lines and gene expression”).

[Editors' note: further revisions were requested prior to acceptance, as described below.]

Essential revisions:1) The current application of the method is thought to be of high value and provides significant biological insight. The method itself is too conceptually similar to earlier work from the Schütz lab (which you have cited correctly) to be referred to as a "novel" assay and thus it is requested that you remove the word "novel" from the title and anywhere else in the manuscript this is claimed, unless you specifically identify the novel aspect. The reviewers acknowledge that you have literally created a "two-hybrid" version based on using the epitope tagged bait (hybrid 1) and the fluorescent tagged prey (hybrid 2), which Schütz didn't fully realise, but they had already used the bait-prey terminology in their 2010 Jove paper, and conceptually it is too small an advance in terms of assay technology. They didn't need to make a first hybrid as they tested a transmembrane bait interacting with a cytoplasmic prey. But they have everything else you describe and more sophisticated analysis.

This is for us the first point of confusion. Perhaps the Editor would like to explain. In the comments on our original submission, the Editors wrote:

"The reviewers agreed that the use of micro contact printing to evaluate lateral interactions MHC class I proteins in situ was novel and presented a useful new approach to such problems." Based on this statement, we assumed that the use of the word 'novel' was supported by reviewers and Editors. Does the more recent comment now mirror the opinion of additional reviewers? Or did the original reviewers change their minds?

In our own opinion, the novelty of our approach is threefold: first, the simple technique of the antibody micropatterns, which in principle allows the method to be adapted by any molecular immunology or cell biology laboratory; second, the ability to investigate, by use of the anti-tag antibody, a defined conformation of MHC class I (and not just any and all MHC class I protein; we are not aware of any other technique that can differentiate different protein conformations in such a clear fashion); and third, the use of the technology to identify and characterize a previously uncharacterized (of course, class I homotypic interactions were previously found in the Zuñiga paper, but it was not possible to her and her coworkers to determine that only the heavy chains were interacting; see the Discussion) interaction (to our knowledge, all previous applications of micropatterns were just proof-of-principle applications, common in biophysical publications, of interactions that were already well-characterized by cell biologists).

We have left the word 'novel' standing in the manuscript for now, but we will not insist on it if the Editor believes that novelty does not apply. We think that upon publication, the quality and novelty of the work will speak for itself; of course, we think that an article will attract more readers and ultimately citations if the word 'novel' is used wherever justified.

2) While the reviewers appreciate that you define cluster in the context of this study, "cluster" with respect to MHC class I antigens has been previously defined and this needs to be respected within papers in this field to avoid confusion. It is also felt that with the ability to detect only a 10% steady state association with the current method (based on use of 1.1 as a threshold) that you apparently can't detect the earlier defined MHC class I clusters. But you are not saying these don't exist. A compromise would be to refer to the phenomenon you are studying as "induced clusters" throughout to distinguish them from previously described spontaneous clusters of MHC class I trimers, that you neglect due to your detection threshold. It is essential to clarify this or use a term not including "cluster".

2.1) About the use of the word 'clusters'. While we 100% appreciate that confusion must be avoided, we think that the word 'cluster' has no clear definition in the field, but has actually been used in three completely different contexts in the literature. We have seen:

- Lu et al., 2012, who define clusters as protein islands on the cell surface. In their paper, they found that certain pMHC (K^b^+SIINFEKL or K^b^+ SIYRYYGL) are found in distinct areas or “clusters” (200-900 nm in diameter). They conclude that “Our most important finding is that endogenous antigen processing generates intracellular clusters of class I molecules segregated on the basis of their peptide cargo that are maintained for hours after their delivery to the cell surface.”

- Pentcheva and Edidin, 2001 (Pentcheva, T., and M. Edidin. 2001. Clustering of peptide-loaded MHC class I molecules for endoplasmic reticulum export imaged by fluorescence resonance energy transfer. J. Immunol. 166:6625–32. doi:10.4049/JIMMUNOL.166.11.6625), who see associations ('clusters') of A2 in the ER;

- Mocsár et al., 2016, who have seen MHC I-Interleukin Clusters (including all MHC I isotypes: A, B, C).

In the review of our first submission, the following sentence is found:

"Cluster has been used (appropriately) in the past to refer to spontaneously concentrated class I molecules, likely present in membrane domains that limit their ability to diffuse from each other. The use of cluster in the present study muddies the waters considerably, and will likely increase confusion, rather than the opposite." We understand from this that the reviewer(s) refer(s) to the Edidin clusters, which are speculated to have something to do with lipid environments. If that is not the case, we would ask the Editor to point out to us the definition of 'clusters' in the literature that they find primary and conclusive. Perhaps, for clarification, it would have been useful if we had been able to see the original remarks of the reviewer.

In addition to the three different definitions of MHC clusters in the literature, the word 'cluster' is also a commonsense word in the English language, and this is the sense that we would have liked to use. It is important to use a commonsense word since people use search engines such as PubMed to find scientific work, and the words that are used to describe phenomena must be commonsense, so that people will be able to identify our paper in *eLife*. This speaks strongly against the use of unusual or invented terms.

Finally the Editor suggests the term 'induced clusters', which we find not fitting, since there is nothing induced or inducible in our associations of class I molecules. One possibility, from our point of view, is 'free heavy chain clusters', which also clearly differentiates from the Edidin work.

The Editor is, and the reviewers are, invited to suggest an alternative wording that is commonsense, correct, and findable. For now, we have left the word 'cluster' standing in the manuscript.

It is also felt that with the ability to detect only a 10% steady state association with the current method (based on use of 1.1 as a threshold) that you apparently can't detect the earlier defined MHC class I clusters.

2.2) It is unclear to us how the reviewer arrived at these two statements, namely that s/he believes that the steady-state association that we show is only 10% (of what?) and that s/he believes that we are somehow not detecting other associations that are described in the literature. We are puzzled as to how a 1.1 threshold would correlate with a '10% steady-state association'. Again, perhaps it would have been helpful if we had been able to read the reviewer's original comment, since there was clearly a misunderstanding which must now be addressed.

In the following, we describe again our method for quantifying the conglomerates as already detailed in our Materials and methods section and in the earlier Smallpaper. To compare the spatial distribution of K^b^-GFP between the pattern elements and the pattern element interspaces, we proceeded as follows:

1) We determined, for each image, a) the mean fluorescence intensity of the entire cell area, and b) the mean fluorescence intensity of the areas on the pattern elements. Quantification was done with ImageJ (National Institutes of Health, Bethesda, USA).

2) We calculated, for each image, the ratio of these numbers of the mean fluorescence intensity on pattern elements over total fluorescence intensity. This means: a theoretical ratio of 1.0 describes a homogeneous distribution of proteins where the pattern elements and the interspaces have the same mean fluorescence intensity.

3) We then checked this ratio in our control experiments, where no clustering was expected, and it was 1.1. This value was thus defined as the threshold.

4) In the experiments, a clear redistribution of proteins onto the pattern elements corresponds to a ratio of 1.3, with the appropriate significance.

With this type of evaluation, we were not at all aiming to determine the percentage of interacting proteins, nor does this analysis allow us to derive them.

It is also felt that with the ability to detect only a 10% steady state association with the current method (based on use of 1.1 as a threshold) that you apparently can't detect the earlier defined MHC class I clusters. But you are not saying these don't exist.

2.3) We think that the reviewer is referring to the MHC class I clusters defined by Yewdell or Edidin (see 2.1.). The fundamental difference between our clusters and these is the following: our clusters are clearly made up of free heavy chains, whereas theirs are associations of trimers (trimer: complex of heavy chain, light chain, and peptide) or mixed associations of trimers and heavy chain/β_2_m dimers.

We do not say that the Yewdell or Edidin clusters do not exist, since we are not in a position to detect them. We are using TAP-deficient cells, in which very few peptides are available to the class I molecules, and so clusters that consist of, or that contain, trimers are not visible in our experimental system. It is important for the reviewer to appreciate this difference. We have now added a sentence in the Discussion that clearly differentiates our clusters from the Yewdell and Edidin clusters.

A compromise would be to refer to the phenomenon you are studying as "induced clusters" throughout to distinguish them from previously described spontaneous clusters of MHC class I trimers, that you neglect due to your detection threshold.

2.4.With respect to the terms 'cluster' and 'induced', please see our statements above in 2.1.

With respect to the 'detection threshold', please see our statements above in 2.2.

3) Proposed new title: "A two-hybrid antibody micropattern assay reveals conformation-specific cell surface induced clustering of MHC I proteins".

We think that it is not prudent to use the word 'induced' (see above 2.1.). One possible similar title would be "A novel two-hybrid antibody micropattern assay reveals cell surface clustering of MHC I heavy chains". We have changed the title of the manuscript accordingly.

4) You describe the quantitative threshold of 1.1 in the response to reviewers, but we couldn't find this in the paper. Please include in the Materials and methods the 1.1 threshold for defining an induced cluster in the system.

The method was referred to from the Small paper (Dirscherl et al., 2017). We have now added the definition and explanation to the Materials and methods in this manuscript as requested.

It is not clear how the gap between 1.1 and 1.3 is defined as a grey zone in the rebuttal, but you should also mention this if you feel it is important to actually set the threshold at 1.3, although this is very close to your experimental result.

We have now added the explanation to the Materials and methods.

[Editors' note: further revisions were requested prior to acceptance, as described below.]

There is no editorial policy against use of "novel" in titles of eLife papers, but this has been infrequent – at around 0.2%. In the first decision letter the editors did override one reviewer in acknowledging that the approach was novel, but when the reviewer persisted in questioning this an editor went to the primary literature and looked at the precedents specifically to address this issue independently. As described in the second decision letter the opinion of the editors was then changed to agree with the reviewer that the core of the assay technology was well described earlier and that this assay is a novel application of an existing assay concept. Since this is too much information to convey in the title, the decision was to remove "novel" from the title and then explain the more nuanced novelty of this application in the text. The editors still ask that you remove novel from the title.

We have now removed the term “novel” from our title.

The second issue has to do with the definition of cluster and whether this is misleading in this context. The editors still ask that you change this terminology to avoid confusion and quantitative implications of the term. The interaction leading to a signal in this assay could be sub-stoichiometric, 1:1 or super-stiochiometric. The term cluster suggests groupings of greater than 1:1, because 1:1 would be called a dimer, and less than 1:1 would not be cluster. This method doesn't provide any information about stoichiometry, but just that it leads to co-localization of prey with the bait by fluorescence microscopy. The heavy chain interaction you are detecting may very well be a dimer, based on other literature, but this cannot be determined from this data. So describing the process as a cluster seems to be an over-interpretation. The bait and pray are simply co-localized at conventional visible light resolution. The authors should refer to co-localization, co-capture or enrichment to precisely describe what the assay detects as a ratio of on/off pattern as quantified in Figure 3?

We have now changed our manuscript and avoid the term 'cluster' as requested. Instead, we describe our findings with the terms “*in cis* (protein-protein) interactions”, “associated” or “co-captured” MHC I proteins. When referring to the findings of other groups, we use the terms that they have used to describe their observations, including the term 'clusters'.

The last issue was the baseline co-localization of prey with bait, which is described as generating a ratio of 1.1 of on/off pattern. This ratio was 1.3 for conditions leading to the empty MHC I heavy chains. Is 1.1 measured under conditions favoring the full heterotrimer of MHC I, peptide and β_2_m, significantly different than 1.0? The quantification was new data added by the authors to Figure 3. This is where 10% came from. The authors may not have data to determine if 1.1 is significantly different than 1.0, but as a discussion point, they might discuss that the previously describe interactions of MHC I proteins on cells may be captured in this number, which would then be a biological background for this assay method and thus will impact its sensitivity. This would also put the relative magnitude of other forms of MHC clustering and this phenomenon in context. More experiments would be needed to sort this out- probably use of different controls to find proteins that are totally unaffected by capture of some of the MHC I molecules to the patterned antibodies. The authors are urged to put these aspects of this assay and earlier investigations of MHC I distribution on the cell surface in context as best they can.

We have now extended our discussions and included the points raised by the Editors.